

# Determining population structure from k-mer frequencies

Yana Hrytsenko[1], Noah M. Daniels[1] and Rachel S. Schwartz[2]

[1] Department of Computer Science and Statistics, University of Rhode Island, Kingston, RI, United States of America

[2] Department of Biological Sciences, University of Rhode Island, Kingston, RI, United States of America

## ABSTRACT

**Background**. Understanding population structure within species provides information on connections among different populations and how they evolve over time. This knowledge is important for studies ranging from evolutionary biology to large-scale variant-trait association studies. Current approaches to determining population structure include model-based approaches, statistical approaches, and distance-based ancestry inference approaches.

**Methods**. In this work, we identify population structure from DNA sequence data using an alignment-free approach. We use the frequencies of short DNA substrings from across the genome (k-mers) with principal component analysis (PCA). K-mer frequencies can be viewed as a summary statistic of a genome and have the advantage of being easily derived from a genome by counting the number of times a k-mer occurred in a sequence. In contrast, most population structure work employing PCA uses multi-locus genotype data (SNPs, microsatellites, or haplotypes). No genetic assumptions must be met to generate k-mers, whereas current population structure approaches often depend on several genetic assumptions and can require careful selection of ancestry informative markers to identify populations. We compare our k-mer based approach to population structure estimated using SNPs with both empirical and simulated data.

**Results**. In this work, we show that PCA is able to determine population structure just from the frequency of k-mers found in the genome. The application of PCA and a clustering algorithm to k-mer profiles of genomes provides an easy approach to detecting the number and composition of populations (clusters) present in the dataset. Using simulations, we show that results are at least comparable to population structure estimates using SNPs. When using human genomes from populations identified by the 1000 Genomes Project, the results are better than population structure estimates using SNPs from the same samples, and comparable to those found by a model-based approach using genetic markers from larger numbers of samples.

**Conclusions**. This study shows that PCA, together with the clustering algorithm, is able to detect population structure from k-mer frequencies and can separate samples of admixed and non-admixed origin. Using k-mer frequencies to determine population structure has the potential to avoid some challenges of existing methods and may even improve on estimates from small samples.

Corresponding author
Yana Hrytsenko,
yhrytsen@bidmc.harvard.edu

## INTRODUCTION

Population structure is a complex phenomenon influenced by the combined effects of various processes such as geographic and demographic barriers. Samples that are located in close geographic proximity tend to look more genetically similar than those that are more remote (*Bradburd, Ralph & Coop, 2016*). When populations are subdivided, they can evolve as separate lineages experiencing differences in recombination, mutation, genetic drift, demographic history, and natural selection (*Andam et al., 2017*). Population structure is thus typically observed as a systematic difference in allele frequencies among populations due to non-random mating among individuals. Genetic differences within and among populations are examined by studies that investigate changes in frequencies of alleles and genotypes over time (*Clark, 2001*; *Andrews, 2010*; *Okazaki et al., 2021*).

Identification of population structure and gene flow among populations is informative for genetic ancestry and provides information about both demographic history and geographic origins (*Novembre et al., 2008*; *Schulman, 2010*; *Smock & Schwartz, 2020*; *Mills & Rahal, 2021*). For example, gene flow, or a gene transfer from one population to another, is indicative of migration processes (*Choudhuri, 2014*). When individuals of a single population possess recent ancestry from two or more separate sources, this population is considered admixed. Admixed populations contain high levels of genetic diversity that reflect contributions of the intermixture of source populations with different genetic variants (*Boca, Huang & Rosenberg, 2020*).

Understanding gene flow among populations informs a diversity of studies of species (*Ellstrand & Rieseberg, 2016*). For example, studies in population structure across marine species have analyzed connectivity among populations, leading to the establishment of networks of marine protected areas (*Pascual, 2019*; *Shen et al., 2019*). Understanding this connectivity among populations, which entails evaluation of population structure across taxa, is a key factor for the effective design of these networks and preserving biodiversity on a large scale. This knowledge is fundamental for ensuring the long-term survival of ecosystems inhabited by these species (*Zhao et al., 2021*).

Population structure is also an important confounding variable in genome-wide association studies ((GWAS) studies that identify genetic variants linked to traits or diseases by scanning the entire genome) due to the possibility of inaccurate associations between genotype and the trait of interest in a genetic study (*Bayless, Brown & Paige, 2017*). The presence of population structure may cause false positive or negative associations between genotype and trait due to differences in local ancestry that are not related to disease risk or trait variance (*Hellwege et al., 2017*). Thus, identification of population structure, and controlling for it, removes the confounding factors (*Li et al., 2010*). This step enables GWAS to find new genetic associations and improve the detection, treatment, and prevention of certain diseases (*Mulcare et al., 2004*; *Ingram et al., 2007*; *Tishkoff et al., 2007*; *Friedrich et al., 2012*).

Three types of population inference approaches are typically used: model-based, distance-based, and statistical (sometimes referred to as algorithmic) (*Alexander, Novembre & Lange, 2009*). An example of a model-based population inference approach is Structured

Association, which assigns samples to subpopulation clusters (possibly allowing fractional cluster membership) using a model-based clustering program such as *STRUCTURE* (*Pritchard, Stephens & Donnelly, 2000*; *Rosenberg et al., 2002*). However, the applicability of this approach to large genome-wide data sets is limited by its high computational cost when allowing fractional cluster membership (*Price et al., 2010*). Faster model-based approaches, such as *admixture* (*Alexander, Novembre & Lange, 2009*), *fastStructure* (*Raj, Stephens & Pritchard, 2014*), and *frappe* (*Tang et al., 2005*) adopt the likelihood model embedded in *STRUCTURE* but incorporate relaxation methods for improving computational efficiency.

However, because these approaches are based on genetic assumptions about the data, including Hardy-Weinberg equilibrium (HWE) within populations and linkage equilibrium (LE) between loci, violating these assumptions may lead to misleading results (*Pritchard, Stephens & Donnelly, 2000*; *Tang et al., 2005*; *Liu & Zhao, 2006*). Marker deviation from equilibrium can also signify a possible sequencing (*Glenn, 2011*; *Mardis, 2013*; *Goodwin, McPherson & McCombie, 2016*; *Salk, Schmitt & Loeb, 2018*) or genotyping error and thus such markers should be excluded from further analysis (*Alhusain & Hafez, 2018*). Incorrect inference of genotypes is known to occur due to low coverage of DNA sequencing (*Lachance, 2016*). The assumptions of Hardy-Weinberg proportions, which must be met for the marker to be included in the analysis, sometimes are not met due to genotyping errors (*Laurie et al., 2010*). Markers that do not meet this requirement should not be included in the analysis (*Alhusain & Hafez, 2018*).

Thus, marker genotypes of single-nucleotide polymorphisms (SNPs) or microsatellites, or haplotype frequencies generated from the sequence data, require careful data preprocessing steps (*Alhusain & Hafez, 2018*). Additionally, before running these methods, the number of populations (K) must be set but may not be known in advance. While model-based approaches are very powerful in population structure identification, they are thus limited by computational cost, operate on genetic assumptions that must be held, and are sensitive to sample size (*Gao & Starmer, 2008*).

Alternative distance-based population inference approaches adopt a pairwise distance matrix computed among each pair of individuals. Some examples of implemented distance-based approaches are genetic similarity score matching (GSM) (*Guan et al., 2009*), Spectral-GEM (*Lee et al., 2010*), and FastPop (*Li et al., 2016*). GSM and Spectral-GEM require high computational intensity when the sample size is large. FastPop results in complex computation and has not been established when inferring genetic ancestry among more than four populations (*Byun et al., 2017*).

Statistical approaches, such as principal component analysis (PCA), which is a linear dimensionality reduction technique, can be applied to genotyped data (individual allele frequencies, SNPs) to extract linear combinations of individuals that share the greatest similarities. A graphical overview (scatter plot) of the population structure can be shown using principal components as axes of variations. PCA is efficient and has been implemented for ancestry inference in *eigenstrat* (*Rosenberg et al., 2002*) and *smartpca* (*Zielezinski et al., 2017*). Statistical approaches are able to handle large-scale genomic datasets and are not restricted by genetic assumptions (*Alhusain & Hafez, 2018*).

PCA has advantages over the model-based approaches in that it is a non-parametric method (it does not require a predefined number of populations) and does not rely on modeling assumptions (HWE, LE). PCA is also computationally efficient. Albeit, current PCA-based approaches, just as model-based approaches, operate on genomic markers, which require careful identification to be useful for population structure analysis.

Overall, methods for identifying population structure tend to be computationally demanding. The number of available markers grows as the number of samples included in the analysis increases, thus reducing the efficiency of computation. Identification of these genotypes requires rigorous steps, and reducing the number of informative markers is often desirable for efficient population structure determination (*Stevens et al., 2011*). Ancestry informative markers are usually determined as a set of minimum markers needed to determine the population structure and lower the genotyping cost. Selection of informative markers using the supervised method relies on self-reported ancestry information from individuals, while the unsupervised approach applies PCA to determine markers that are associated with the significant principal components and then score each marker (*Paschou et al., 2007*).

In this work, we investigate our ability to determine population structure with PCA using frequencies of k-mers present in a genome. More specifically, we examine the ability to differentiate the population structure using samples across five superpopulations identified by the 1000 Genome Project (*The 1000 Genomes Project Consortium et al., 2015*). It is important to note that in this study "superpopulation" refers to a broad categorization of human populations often used in genetic analyses to represent groups with shared genetic patterns, typically influenced by geographic proximity and historical migration. This grouping of genetic diversity into categories often reflects the structure of sampling strategies rather than discrete boundaries in genetic variation (*Rosenberg et al., 2002*; *Serre & Paabo, 2004*; *Altshuler, Donnelly & The International HapMap Consortium, 2005*; *The 1000 Genomes Project Consortium et al., 2015*). Information on the presence and absence of k-mers has shown promising results in population differentiation whole-genome sequencing reads from two distinct superpopulations (*Rahman et al., 2018*). K-mers are shorter substrings that can be "overlapped" to reconstruct the full sequence and deliver equivalent genomic information as a whole sequence (*Compeau, Pevzner & Tesler, 2011*). K-mers or k-mer profiles of a sequence (k-mer and its frequency in the genome) can be generated efficiently (*Marcais & Kingsford, 2011*). Then the structure of the genome can be investigated directly from the k-mer profiles of the genome. The problem of sequencing or genotyping errors may be reduced because PCA aggregates the k-mer frequency information; therefore, extra counts of frequencies that could potentially accrue due to sequencing errors should not substantially influence the PCA projection. Additionally, sequencer errors can be identified and removed by filtering out k-mers of frequency one (singletons), which are generally considered a result of sequencer errors (*Shendure & Ji, 2008*; *Wang et al., 2019*), and not to be included in the final PCA computation.

In this study, we examined the ability to determine population structure based on k-mer frequencies present in a genome, with the goal of formulating a quick and accurate approach for population structure identification based on data that provides an alternative

approach and information from typical markers. To compare our approach to existing methods, we used samples from across human populations from the 1000 Human Genomes Project (*The 1000 Genomes Project Consortium et al., 2015*). The structure of these populations has been established previously and includes both the separation of the five superpopulations, independent populations within these, and populations with mixed ancestry. We were able to apply PCA to frequencies of k-mers present in genomes to accurately separate superpopulations and populations, and identify individuals of mixed ancestry. Additionally, we were able to confirm the use of this approach, and its sensitivity to some evolutionary parameters, using simulations. Using simulated data, we demonstrate that the results produced by our approach of using k-mer frequencies and PCA with K-means are comparable to those produced by the widely-used, model-based method, *fastStructure*, that operates on SNPs data. We show that our approach is able to identify populations using sample k-mer frequencies, even in the scenarios where *fastStructure* assigns samples from two different populations to a single population due to the presence of highly admixed samples.

For comparison, we investigated population stratification based on the number of k-mer matches between pairs of genomes using a popular alignment-free sequence comparison tool, *mash* (*Ondov et al., 2016*). We were able to build accurate population trees using this approach; however, the results depended on the parameter selection and it was difficult to identify *a priori* the k value (k-mer length) and sketch (reduced representation of a sequence) size needed for accurate results. Thus, the practicality of this approach is limited compared to our PCAs of k-mer frequencies.

Portions of this text were previously published as part of a preprint (DOI 10.21203/rs.3.rs-1689838/v2).

## MATERIALS AND METHODS

In order to examine our ability to determine population structure based on k-mer frequencies present in a genome, we obtained genome sequence data from samples across human populations. Human population structure has been studied extensively, suggesting that humans consist of five superpopulations (*The 1000 Genomes Project Consortium et al., 2015*) (Africa, America, Europe, South Asia, East Asia). These groups or superpopulations, which correspond loosely to geographical regions, are not biologically distinct categories but reflect patterns of genetic similarity influenced by geography and sampling. Furthermore, the genetic diversity we observe today is shaped by contributions from multiple ancestral populations rather than discrete, original human groups (*Rosenberg et al., 2002*; *Serre & Paabo, 2004*; *Altshuler, Donnelly & The International HapMap Consortium, 2005*). Within each of these superpopulations are multiple populations generally corresponding to geography. By examining samples whose clustering is well established we are able to determine how a novel method performs compared to existing approaches. Specifically, we compare our results to those of the 1000 Genomes Project (*The 1000 Genomes Project Consortium et al., 2015*), which provides a "gold standard" for population structure based on SNPs using the program *STRUCTURE*. We suggest that if the ability of our novel

approach compares favorably to known structure in an exemplary dataset, we can consider how to apply this to other systems.

Our initial comparison was designed to be the set of individuals that should be the easiest to differentiate: we selected six samples from a single population from each of the five major superpopulations:

1. Luhya population in Webuye, Kenya (LWK) of African ancestry.
2. Peruvian population in Lima, Peru (PEL) of American ancestry.
3. Toscani population in Italy (TSI) of European ancestry.
4. Indian Telugu population in the UK (ITU) of South Asian ancestry.
5. The Japanese population in Tokyo, Japan (JPT) of East Asian ancestry.

All samples had been sequenced using polymerase chain reaction (PCR)-free high coverage technology and listed under the "1000 Genomes 30x on GRCh38" data collection (see Data availability section). Data was accessed as *cram* files. Each *cram* file was converted into a *bam* file using samtools (version 1.12) (*Li et al., 2009*). We then used the bcftools (version 1.12) (*Danecek et al., 2021*) mpileup command to filter out regions with low-quality scores, call the variants, and perform pileups. Finally, the *fasta* files were built using the bcftools consensus command. While k-mers can be efficiently counted directly from raw sequencing reads, we chose to count k-mers from whole genome sequences (WGS) to ensure higher accuracy by eliminating sequencing errors and artifacts common in raw reads. WGS provide a more complete representation of the genome, reducing redundancy and enabling more consistent comparisons across samples. Additionally, using WGS captures the full genomic context, including repetitive regions, which is important because repetitive regions make up a significant portion of many genomes. Sequencing reads may fragment or underrepresent these regions due to technical challenges, such as mapping difficulties or low coverage.

## Population structure of human superpopulations from k-mer frequencies using PCA

For each sample, we computed the frequencies of canonical k-mers (k-mer or its reverse complement, whichever comes first lexicographically) of length 21 bp. Frequencies were computed from the *fasta* file using Jellyfish (*Marcais & Kingsford, 2011*) (version 2.2.10), a tool for fast, memory-efficient counting of k-mers in the DNA sequence. When choosing the length of k-mers we were initially guided by a general rule used by alignment-free methods for sequence comparisons—shorter k-mers are more likely to be present in a sequence (*e.g.*, 1-mers); thus they are less informative in analyzing closely related genomes (*Bernard, Chan & Ragan, 2016*); however, longer k-mers are more unique to particular species and are therefore more useful for similarity identification across species (*Greenfield & Röhm, 2013*). Moreover, widely used alignment-free sequence comparison tools such as *mash* found a k-mer length of 21 to give accurate estimates of sequence similarities (*Ondov et al., 2016*). We examined alternate values of k (k-mer frequency length) in our simulations (see below).

We filtered out singletons (k-mers of frequency one) to account for possible errors produced by the sequencer (*Melsted & Pritchard, 2011*). For this purpose, the flag -L 2 is

used when analyzing the k-mer content of the sequence with Jellyfish. We then sorted each k-mer frequency profile in alphabetical order and calculated the intersection of k-mer profiles across all samples. We used the intersection to ensure accurate counts, as singletons may be sequencing errors and were not counted; thus, including such k-mers could introduce the uncertainty of assigning counts of either 1 or 0. The resulting output is a count of each k-mer of size 21 that is found in all samples.

We performed a PCA on vectors of k-mer frequencies of each sample in *python* (version 3.6.6) using the scikit-learn (sklearn) library (*Pedregosa et al., 2011*) (version 0.23.2). We normalized the data (frequencies of k-mers) by scaling all the values to be between 0 and 1 using the StandardScaler function from the scikit-learn (sklearn) library (*Fränti & Sieranoja, 2018*). Normalization of the dataset is a necessary step due to PCA calculating a new projection of the dataset with new axes based on the standard deviation of the variables (*Jolliffe & Cadima, 2016*). Variables with different standard deviations (high *versus* low) will have different weights for axes calculation. Normalization of data allows for uniform standard deviation across all variables, thus PCA calculates axes with all variables having equal weight. We visualized the projection of the two PCs with the most variance using a scatter plot from the matplotlib package (*Hunter, 2007*).

To identify populations, we used K-means clustering based on the PCs (*Ding & He, 2004*; *Lee, Abdool & Huang, 2009*) using the scikit-learn (sklearn) library (version 0.23.2). We found the optimal number of principal components that capture the greatest amount of variance in the data by plotting the explained variances in a scree plot. We used the explained variance ratio as a metric to evaluate the usefulness of the principal components and to choose how many components to use in the model (*Jombart, Devillard & Balloux, 2010*). The explained variance ratio is the percentage of variance that is attributed by each of the selected components. To avoid overfitting the model we chose the number of components to include in the model by adding the explained variance ratio of each component until we reach a total of around 80%, which is considered an adequate amount of variance to derive informative results (*Jolliffe & Cadima, 2016*). We used these PCs to cluster samples into different numbers of groups (k from 1–10). We determined the optimal number of clusters (K) by using the "elbow method" heuristic approach (*Yuan & Yang, 2019*). The sum of the squared distances to the nearest cluster center (aka *inertia*) was measured using the K-means model for each k. From the scree plot, we determined the "elbow point", *i.e.,* the point after which the inertia starts decreasing in a linear fashion. We clustered the dataset by fitting the numbers of PCs with 80% of the variance into the K-means model with k number of clusters.

Because the PCs with 80% of the variance caused undecidability in the K-means clustering algorithm, we hypothesized that the dataset contains noise that suppresses the true biological signal, and only a small fraction of k-mers "drive" (dominate) the data. From the scree plots of PC variance and K-means inertia plots, we saw that the variance is spread out, mostly equally, through all the PCs. Commonly, the first three PCs contain most of the variance of the data (around 80%), however, our dataset shows that the first three PCs have only a slightly higher variance than the rest of the PCs. Nevertheless, when

using the first two PCs we saw deterministic results by K-means and they matched the expected results. Thus, we used the first two PCs throughout this work.

The analysis of the dataset including all five superpopulations showed strong differentiation of the AFR superpopulation from the rest of the superpopulations. Thus, we repeated the analysis for four superpopulations excluding AFR, with 24 samples of a single origin as identified by *The 1000 Genomes Project Consortium et al. (2015)* (AMR_PEL, EAS_JPT, EUR_TSI, and SAS_ITU).

## Differentiating human populations with k-mer frequencies using PCA

Because we were able to differentiate human superpopulations, we examined whether this approach could differentiate populations within those superpopulations. We obtained an additional 12 samples (six per population) of European ancestry and East Asian ancestry originating from single-origin populations (*The 1000 Genomes Project Consortium et al., 2015*):

6.  Finnish population in Finland (FIN) of European ancestry.
7.  Dai Chinese population in Xishuangbanna, China (CDX) of East Asian ancestry.

Genomes for each individual were obtained as described above. We repeated our analysis with these six populations from four superpopulations (excluding AFR). These additional samples allow us to establish how this approach differentiates samples hierarchically (*i.e.,* whether all six populations are differentiated or whether only the four superpopulations are supported). Methods such as *STRUCTURE* often require separate examination of individual clusters to determine large-scale and fine-scale population structure separately (*O'Neill et al., 2013*).

## Populations from multiple ancestral origins

To determine whether we could identify populations of more complex origin, we selected samples from populations that were previously identified as comprising roughly equal parts fractional membership of two ancestral populations (*The 1000 Genomes Project Consortium et al., 2015*) from the same superpopulation (we refer to these as "multiple-origin" populations). We obtained six samples from the Han Chinese population in Beijing China of mixed East Asian ancestry (CHB). In our prior analysis the two ancestral populations within East Asia are represented by JPT and CDX. We obtained an additional six samples from Utah residents of Northern and Western European ancestry (CEU). In our prior analysis Northern European ancestry is represented by the FIN population, while Western European ancestry is represented by TSI. Genomes for each individual were obtained as described above. We repeated our analysis with these eight populations from four superpopulations (AMR_PEL, EAS_CDX, EAS_CHB, EAS_JPT, EUR_CEU, EUR_FIN, EUR_TSI, and SAS_ITU; excluding AFR).

Because we continued to differentiate the four superpopulations, we focused a subsequent analysis on the single East Asian (EAS) superpopulation with 18 samples (12 single-origin and six multiple-origin; EAS_CDX, EAS_CHB, and EAS_JPT) to determine our ability to identify population origin as in *STRUCTURE* (*Pritchard, Stephens & Donnelly, 2000*). We repeated our PCA analysis on these samples alone. Additionally, we repeated the

analysis on the single European (EUR) superpopulation with 18 samples (12 single-origin and six multiple-origin; EUR_CEU, EUR_FIN, EUR_TSI).

## Comparison of k-mer/PCA results to SNP/fastSTRUCTURE

To compare our k-mer frequency-based PCA approach, we assessed population identifiability in the same datasets from the 1000 Genomes Project using SNPs and *fastStructure* (*Raj, Stephens & Pritchard, 2014*, version 1.0). This tool is a computationally efficient alternative to the *STRUCTURE* method (*Pritchard, Stephens & Donnelly, 2000*). We extracted SNPs for each group of samples using bcftools (version 1.12) to merge the vcf files generated previously from cram files. Data was then converted to bed format using PLINK (version 2.00a3.7). fastStructure was run with K values ranging from two to six for all datasets except in the case where there were eight potential populations. The *chooseK* script included in *fastStructure* was employed to determine the optimal K values. Finally, the *distruct* script included in *fastStructure* was used to create plots for the selected optimal K value(s).

## K-mer frequency approach applied to simulated data

To further investigate the use of k-mer based methods in population structure analyses, we applied the same approach to simulated data, where the true population structure was known. We initially simulated three populations using a genome size of $10^7$ bp. Simulations were performed in SLiM (*Haller & Messer, 2017*) in the GUI (SliMguiLegacy version 3.7.1) using a random starting sequence, neutral mutations at a rate of $10^{-7}$ and a recombination rate of $10^{-8}$. The initial effective population size of the starting population was set to 500, and this population was expanded to 1,000 after 2,000 generations. A second population of 180 was added in generation 3,500, and a third population of 180 was added in generation 4,500. Minimal migration was allowed. Six genomes per population were sampled at generation 7,000. This model was loosely based on the human population model of *Gravel et al. (2011)*, although with fewer generations due to time and computational limitations, and no population expansion.

K-mers were counted as above using Jellyfish (*Marcais & Kingsford, 2011*) (version 2.2.10), a PCA was produced based on the intersection of k-mers across samples, and K-means clustering was used to identify groupings. Because our initial approach using k-mers of size 21 produced no k-mers with counts greater than two, and even fewer k-mers common to all samples, we used k-mers of size 9. K-mer size was chosen by examining k-mer frequency distributions and selecting a size with a symmetrical distribution.

The simulation was repeated with genomes sampled at 5,500 generations. This simulation was then repeated with exponential growth in the second and third populations following the establishment of the third population (similar to expected population expansion in humans). Finally, we used the simulation without growth and added a fourth population that was the product of equal migration from populations 2 and 3. After a single generation, this population was isolated.

To examine the impact of hybrid-origin populations more precisely, we adapted a SLiM simulation population structure recipe for adding subpopulations. We started with three

populations of effective populations sizes of 500, 500, and 200. The smaller population experienced 10 generations of equal origin from the larger two. After this, migration was reduced to 0.02% of the smaller population originating from each of the larger populations in each generation (probabilistically) and each of the larger populations experiencing origin of 0.02% from the smaller population. Genomes were sampled after 5,500 generations. We used a k-mer size of 11 for this and subsequent simulations based on k-mer distributions. Simulations were repeated with migration of 0.2% and 2% to examine the impacts of higher migration rates on detection of populations.

## Comparison of simulated data results using fastSTRUCTURE

For comparison, we examined population differentiation in our simulated datasets using *fastStructure* (*Raj, Stephens & Pritchard, 2014*) (version 1.0) as described earlier. We extracted the biallelic sites from the simulated whole genome alignments and converted them to *STRUCTURE* format using a custom script. We ran *fastStructure* for K values of 2–6 and used the *chooseK* script to select the optimal range of K values. We used the *distruct* script to generate plots for the optimal value(s) of K.

## Population structure from a number of shared k-mers between sequence pairs

For comparison with our k-mer frequency-based PCA approach, we examined population identifiability in our human datasets using *mash* (*Ondov et al., 2016*) (version 2.1.1). *mash* can be used to build phylogenies for family-level data and shows promise for population genetic analyses of polyploid sequences (*VanWallendael & Alvarez, 2022*). The principle behind *mash* is that each sequence is converted into a MinHash sketch, a vastly reduced representation of a sequence, then two sketches are compared by calculating the fraction of shared k-mers between a pair of sequences (Jaccard index). Finally, the *mash distance* is calculated, which estimates the rate of sequence mutation under a simple evolutionary model. *mash* has been investigated for basic population genetic analyses of polyploid and diploid species (*VanWallendael & Alvarez, 2022*) and showed some promising results in the population stratification of plants.

For each sample we built *mash* sketches using a *mash* sketch command with -m 2 flag to filter out single k-mers, -k N flag to analyze the k-mer length of *N,* and -s M flag to build sketches of size *M*. We repeated the process for parameters of -k N = 21, 24, 27, 29, 32 and -s M = 1,000, 3,000, 5,000, 7,000, 8,000, 9,000, 10,000, 12,000, 15,000, 18,000, 20,000, 23,000, 25,000, 28,000, 30,000 to compare the results for a different set of parameters. We calculated pairwise distances between each pair of samples using *mash* and used the distances to build a neighbor-joining tree (NJ) for each set of parameters—k-mer length and sketch size. *mash* does not assign samples by population thus to verify that grouping by superpopulation we checked for monophyly of each of the groups in the NJ tree by superpopulation label. We used the *is.monophyletic* function in the ape (version 5.0) R package (*Paradis & Schliep, 2019*) to check whether each population was monophyletic in the resulting tree and *ggplot2* (*Wickham, 2016*) (version 3.3.4) to plot the Boolean values for whether the tree with all the populations contained clades that are all monophyletic.
## RESULTS

### Population structure of five human superpopulations shows two major groups

To differentiate samples from the five human superpopulations identified by the 1000 Genomes Project (*The 1000 Genomes Project Consortium et al., 2015*) we used principal component analysis with a K-means clustering algorithm. 80% of the variance was captured by 21 PCs. The first two PCs contained 14.5% of the variance.

Using 21 PCs, which hold 80% of the variance, caused undecidability in the K-means clustering algorithm where no obvious inflection ("elbow") point was observed (Fig. 1A). However, using the first two PCs, which hold 14.5% of the variance, showed deterministic results in K-means clustering (Fig. 1B). Thus, we applied K-means clustering to the first two PCs throughout this study. The African superpopulation was strongly differentiated from all other samples along PC1. The "elbow point" of the K-means scree plot (when the change in the value of inertia is no longer significant) indicated the samples grouped into three clusters. There was a strong differentiation of samples of African Ancestry (AFR_LWK) from all other populations on PC1, and differentiation within this population along PC2 (Fig. 2). In contrast, the *fastStructure* SNP-based approach assigned the individuals to a single population (Fig. SA1).

### Population structure from four human superpopulations shows four groups

To determine whether we could differentiate the non-African populations we repeated the PCA analysis excluding AFR_LWK samples. 80% of the variance was captured by 18 PCs. The first two PCs contained 13% of the variance. Plotting the first two PCs, we observe four distinct groups corresponding to the four superpopulations (Fig. 3). The elbow point of the K-means scree plot indicated the samples grouped into four clusters (Fig. SA2). In contrast, the *fastStructure* SNP-based approach assigned the individuals to a single population (Fig. SA3).

### Population structure from four human superpopulations including samples of multiple ancestral origin

To examine the ability of this approach to provide information about samples from populations identified as having ancestry from multiple populations within a single superpopulation (as identified by the 1000 Genomes Project), we repeated the PCA analysis with additional "multiple-origin" samples from EAS and EUR superpopulations. 80% of the variance was captured by 36 PCs. The first two PCs contained 8.4% of the variance. Plotting the first two PCs, we observe four distinct groups corresponding to the four superpopulations; individual populations were not clearly differentiated (Fig. 4A). The "elbow point" of the K-means scree plot indicated the samples grouped into four clusters (Fig. SA4). On the other hand, the *fastStructure* SNP-based approach selected a K value between 1–5; however, *fastStructure* plots assigned individuals either to a single population (Fig. 4E), or wholly or partly to one of two populations, which did not correspond

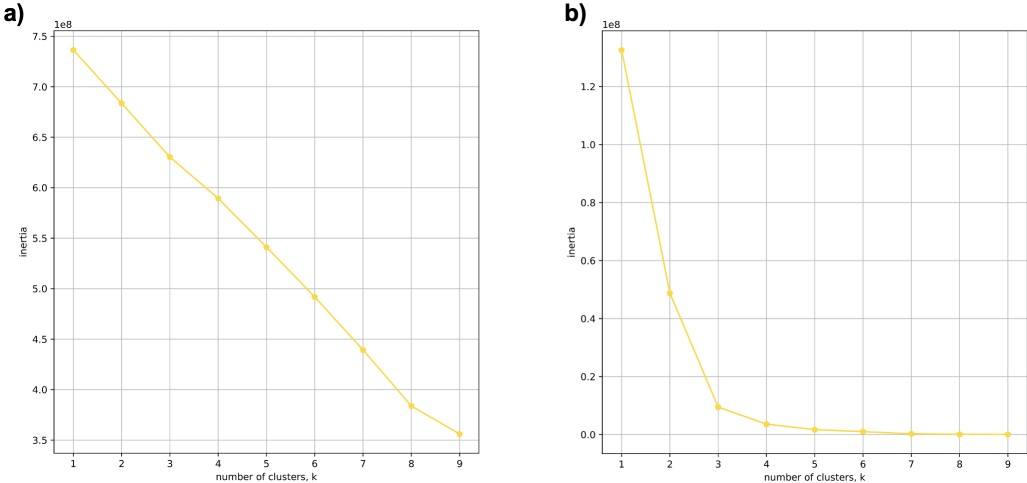

**Figure 1** **Evaluating cluster determination in superpopulations by K-means based on the number of PCs.** Comparison of (A) Scree plot showing the non-deterministic number of clusters (no "elbow point") determined by K-means using 21 PCs (80% of the variance) from k-mer frequencies of five superpopulations. (B) Scree plot showing a deterministic number of clusters = 3 ("elbow point") determined by K-means using 2 PCs (14.5% of the variance) from k-mer frequencies of five superpopulations.

to expected population differentiation or that identified by our k-mer/PCA approach (Figs. 4B–4D).

## Population structure from three populations of single and multiple origins

To further examine our ability to understand populations with samples identified as having multiple origins, we repeated the PCA analysis with our samples from the East Asian superpopulation. 80% of the variance was captured by 14 PCs. The first two PCs contained 13% of the variance. Plotting the first two PCs, we observe three distinct groups corresponding to the three populations CDX, CHB, and JPT (superpopulation EAS), with the CHB population placed in between CDX and JPT populations (Fig. 5). The "elbow point" of the K-means scree plot suggested three populations, although it was difficult to distinguish between three to five, suggesting the samples formed a continuous grouping (Fig. SA5). When samples were assigned to three populations these corresponded to the known populations, with one exception. In contrast, the *fastStructure* SNP-based approach assigned the individuals to a single population (Fig. SA6).

We also repeated the PCA analysis with samples from the European superpopulation with samples of both single and multiple origin. 80% of the variance was captured by 14 PCs. The first two PCs contained 13% of the variance. Plotting the first two PCs, the three populations CEU, FIN, and TSI (superpopulation EUR) appear visually differentiated along PC 1, with CEU (Northern and Western ancestry) placed in between FIN (Northern) and TSI (Western) (Fig. 6). As for the prior analysis, the "elbow point" of the K-means scree plot suggested three populations, although the angle differentiation was not strong, suggesting the samples formed weaker groupings than the superpopulations (Fig. SA7).

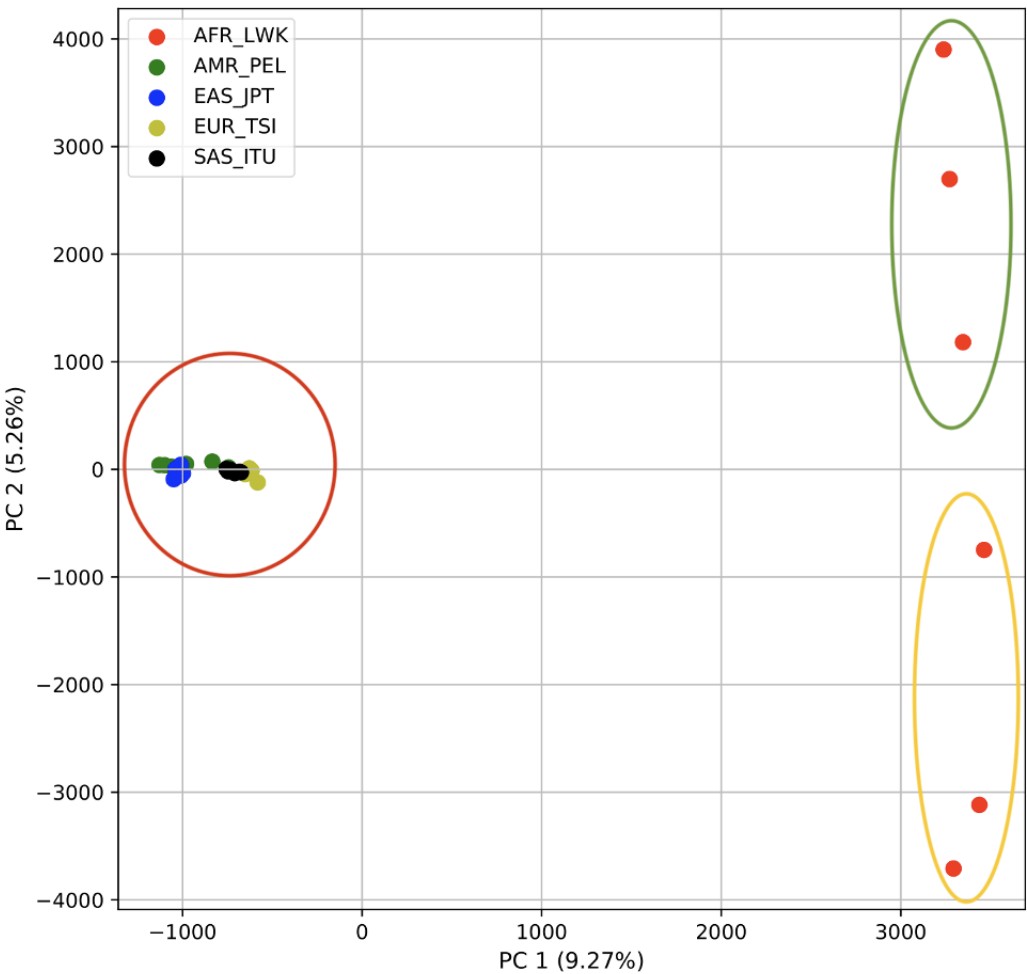

**Figure 2** **PCA of human superpopulations based on k-mers.** PCA generated using k-mer frequencies from a single population from each of five human superpopulations. Samples are colored by population. K-means algorithm identified three clusters (circled) present in the data: two in Africa (AFR) and one including all other populations, Americas (AMR), East Asia (EAS), Europe (EUR), and South Asia (SAS).

While the visual scatter plot of the PCA appears to suggest differentiation along PC 1 nearly consistent with the expected population groupings, there is discordance between sample populations and K-means assignments. This is because K-means clusters individuals into genetically homogeneous subpopulations by placing each observation to the cluster with the nearest mean (*Fränti & Sieranoja, 2018*). On the other hand, the *fastStructure* SNP-based approach assigned the individuals into a single population (Fig. SA8).

## Accuracy of K-means clustering

The adjusted mutual information (AMI) score for K-means clusters and the expected clusters from the analysis of the five superpopulations of non-admixed origin was 0.35 but improved to 1.0 when excluding AFR. The AMI for K-means clusters and the expected clusters of the four superpopulations including samples of admixed and non-admixed origin was 1.0. When analyzing individual superpopulations including samples of admixed

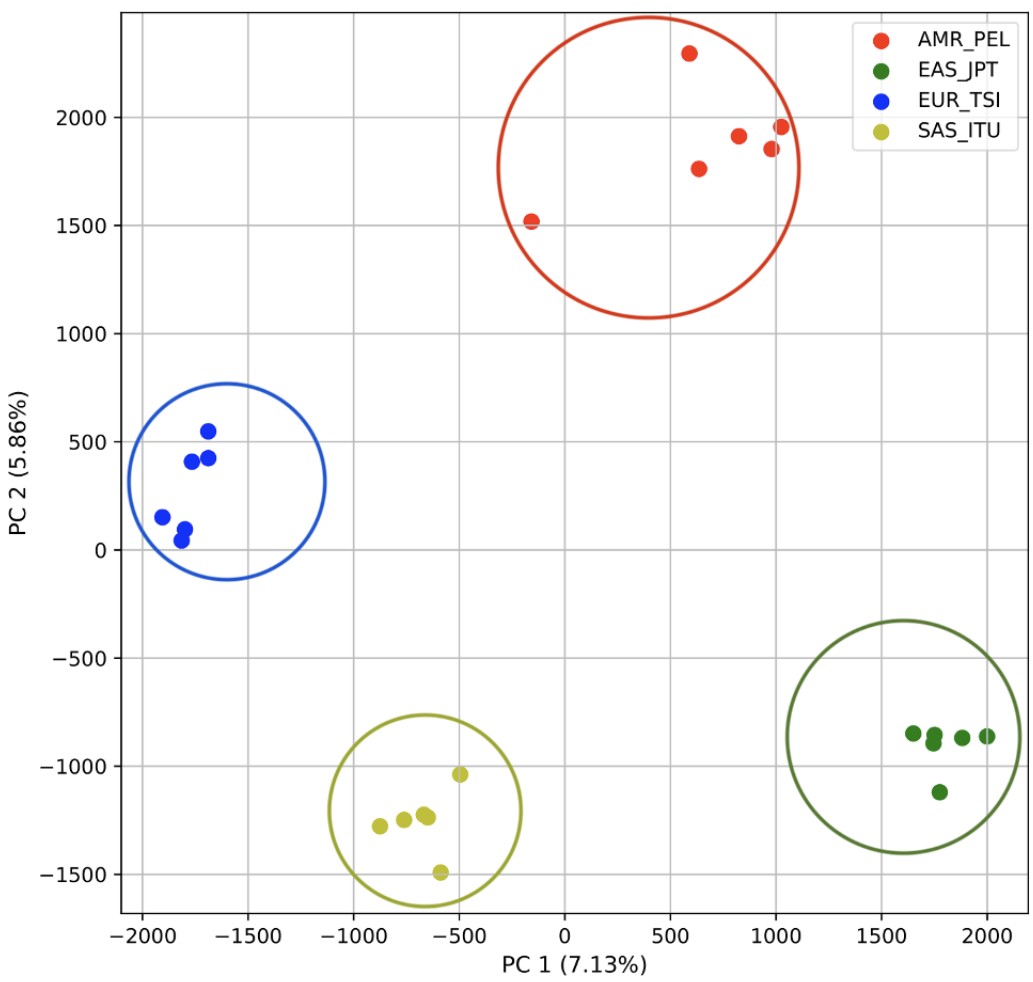

**Figure 3** **Distinct superpopulation clusters in non-admixed superpopulations.** PCA generated using k-mer frequencies from four superpopulations of non-admixed origin (America (AMR), East Asia (EAS), Europe (EUR), South Asia (SAS), but excluding Africa (AFR)) using 2PCs. Samples are colored by population. K-means algorithm identified four clusters present in the data (circled).

and non-admixed origin to differentiate populations, the AMI was 0.83 for the populations in EAS, and 0.43 for the populations in EUR. For the explained variance plots and for the cumulative explained variance plots see Figs. SA9–SA18.

### Memory usage when analyzing population structure with k-mer frequencies

These analyses (using k-mer length of 21) stored a dictionary data structure with k-mer content for each sequence. The dictionary of 48 vectors of k-mers and their frequencies occupied 41Gb of space in a pickled (compressed) format. Reduction of space to 38Gb was possible by calculating the intersection of k-mers across all vectors. Generating these data structures required an HPC node with at least 250Gb of RAM for the dictionary compression step. Additionally, calculating PCA on this dataset required ~239Gb of RAM,
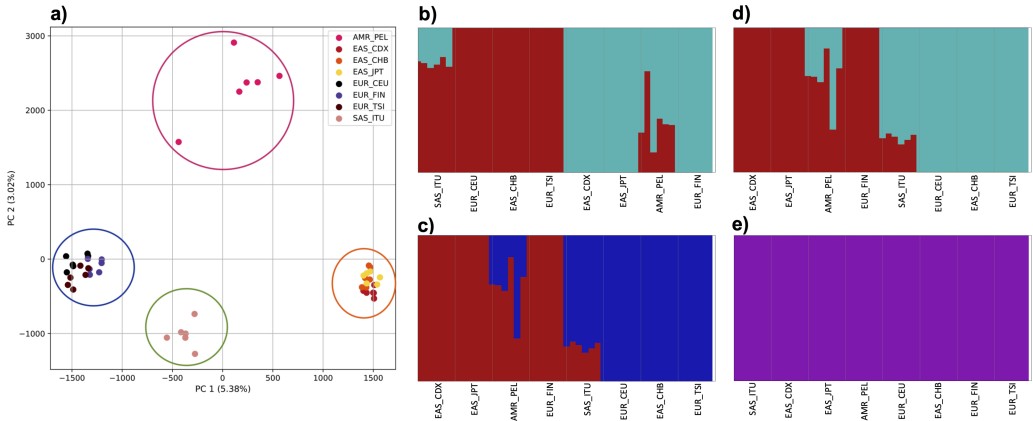

**Figure 4  Identification of distinct clusters on a superpopulation level and a single cluster on a population level.** (A) PCA generated using k-mer frequencies from four superpopulations (America (AMR), East Asia (EAS), Europe (EUR), South Asia (SAS)) including samples of single origin and multiple origin in EAS and EUR using 2PCs. Samples are colored by population. K-Means algorithm identified four clusters present in the data (circled). (B–D) *fastStructure* assignment of individuals to two populations with $K = 2, 3, 4$ respectively determined as optimal K by the *chooseK* method. (E) *fastStructure* assignment of individuals to a single population for $K = 5$.

and took 2 h 36 min and 51 s of job wall-clock time on a 36-core HPC node. For simulations of 10 million base pairs and 18–24 samples, analyses took just a few minutes.

## Accuracy of population structure estimates from k-mer frequencies using simulations

To examine the use of this approach in greater depth we simulated populations and sampled genomes from each. In our initial simple three-population simulation (Fig. 7), we clearly identified the three populations using k-mer frequencies with PCA and K-means clustering. The *fastStructure* SNP-based approach also assigned the individuals in the three populations correctly.

Reducing the time interval between the establishment of the third population and sampling did not affect the results either for our approach (Fig. 8) or from *fastStructure*.

However, when the second and third populations experienced exponential growth, they were grouped together in the PCA (separated from population 1 along PC1) and by K-means clustering, while the oldest population was separated into two groups (along PC2) (Fig. 9). *fastStructure* also grouped the second and third populations, although it did not separate population 1 into two groups.

When these two populations were analyzed without population three, they were separated by the PCA and K-means clustering (Fig. 10). *fastStructure* also separated the populations successfully, but only suggested the two clusters accurately.

For the four populations simulation, where the fourth population originated from a mix of two others, followed by isolation, all four populations were separate, with population 1 differentiated along PC1 and the remaining along PC2 (Fig. 11). In contrast, *fastStructure* supported either two or three populations, but was unable to differentiate

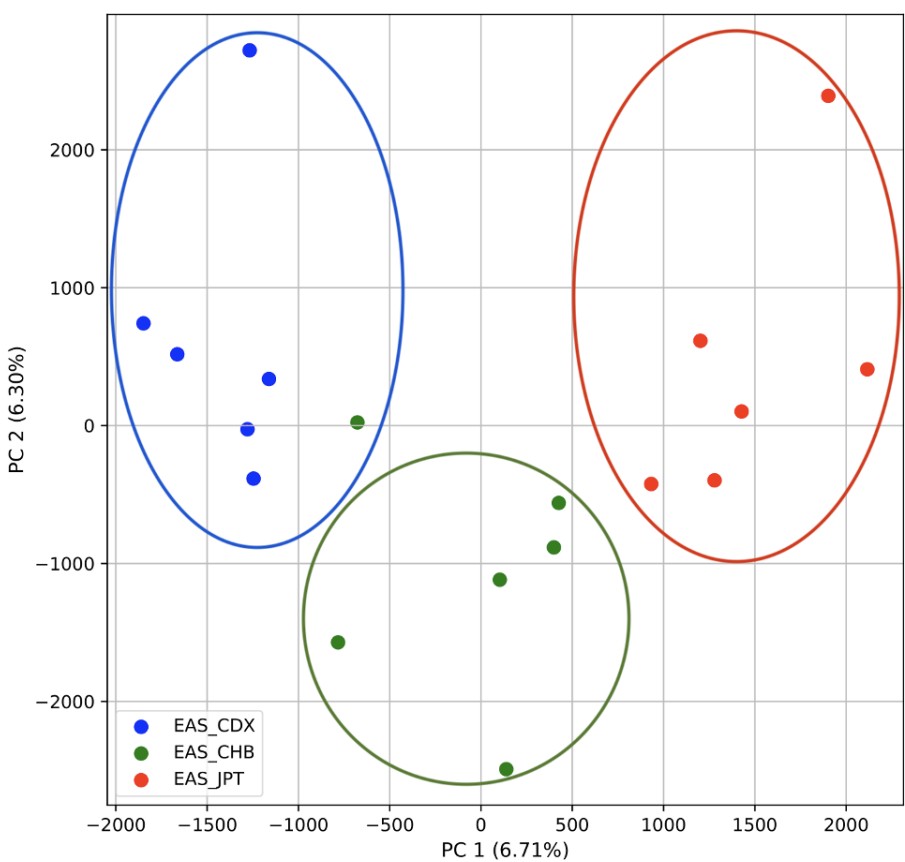

**Figure 5** **Identification of clusters on a population level.** PCA generated using k-mer frequencies from the EAS superpopulation including samples of single and multiple origin (CDX, CHB, and JPT) using 2PCs. Samples are colored by population. K-means algorithm identified three clusters present in the data (circled) corresponding closely to the expected populations.

all four successfully. In the former case, population 1 was separated from all others (*i.e.* matching PC1 of the k-mer approach), while in the latter population 1 and 2 were each separate, while 3 and 4 were grouped together.

In using simulations to examine hybrid-origin and subsequent migration more closely, isolated populations of hybrid origin were rapidly differentiable from other populations (Fig. 12) using both approaches.

As migration increased, populations became more difficult to differentiate. Populations with limited ($\leq 0.2\%$ of parents from each origin population entering the hybrid population and vice versa) were differentiated, but had some overlap and contained one mis-assigned individual in the clustering (Fig. 13). The *fastStructure* approach suggested either three or four clusters. In the 3-population scenario, the first population was strongly differentiated (*i.e.,* matching PC1 of the k-mer approach), with the addition of the partial assignment of an individual from population 3. Five individuals from population 2 were strongly differentiated with the addition of an individual from population 3. The remaining

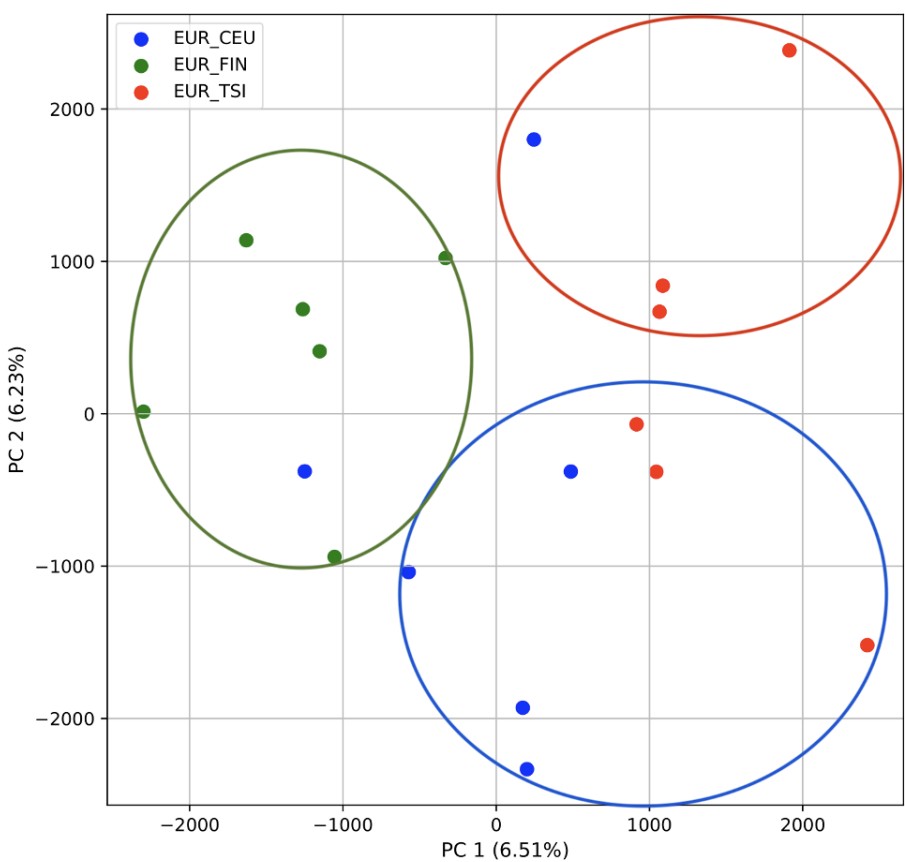

**Figure 6 Identification of clusters on a population level in samples of single and multiple origins.** PCA generated using k-mer frequencies from the EUR superpopulation including samples of single and multiple origin (CEU, FIN, and TSI) using two PCs. Samples are colored by population. K-means algorithm identified three clusters present in the data (circles). Populations appear to separate along PC 1; however, K-means clustering differentiates the two single-origin populations (FIN and TSI) but mixes samples of CEU and TSI, as well as CEU and FIN.

individuals from population 3 were strongly differentiated with the addition of an individual from population 2 and the partial assignment of the individual to population 1.

When divided into four clusters, two individuals from population 1 formed a cluster, the four remaining individuals formed a cluster along with partial assignment of a single individual from population 3. Population 2 formed a cluster combined with one individual from population 3 (as in the k-mer approach), although one individual was only partially assigned, and the five remaining individuals from population 3 formed a cluster.

As migration was increased further, populations could not be differentiated (Fig. 14). While the k-mer approach with K-means identified three clusters, these did not match the simulated populations. Similarly, *fastStructure* identified either two or three populations. In the former scenario, one cluster consisted of two individuals from population 3 and three from population 2, and 50% assignment of an individual from population 1. In the latter, cluster 1 consisted of 5 individuals from population 1, 3 from population 2, and 2

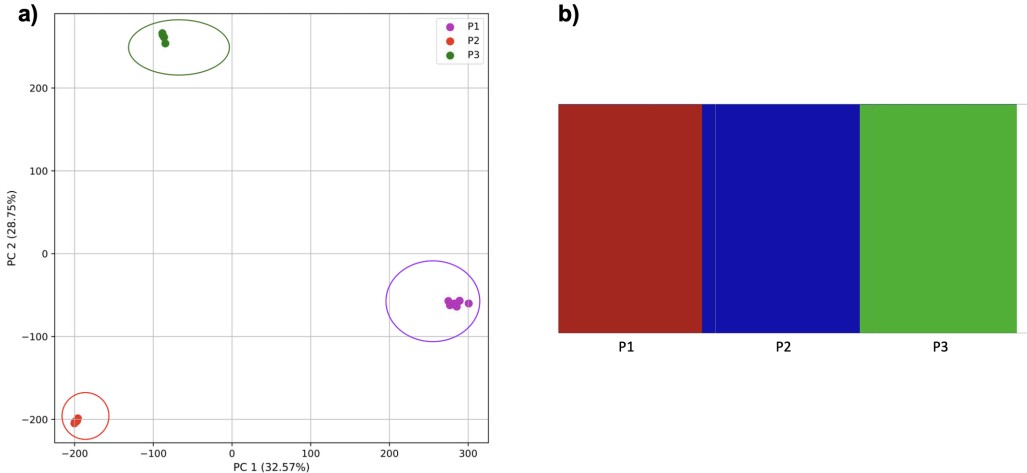

**Figure 7** **Comparison of population stratification approaches using simulated data.** Comparison of population stratification approaches using three simulated populations loosely based on the human out-of-Africa population model of *Gravel et al. (2011)*. (A) PCA generated using k-mer frequencies from samples. Samples are colored by simulated population. K-means algorithm accurately identified three clusters present in the data (circled) corresponding to the three simulated populations. (B) The same three populations accurately identified from SNPs using *fastStructure*. Samples are colored by the assigned population.

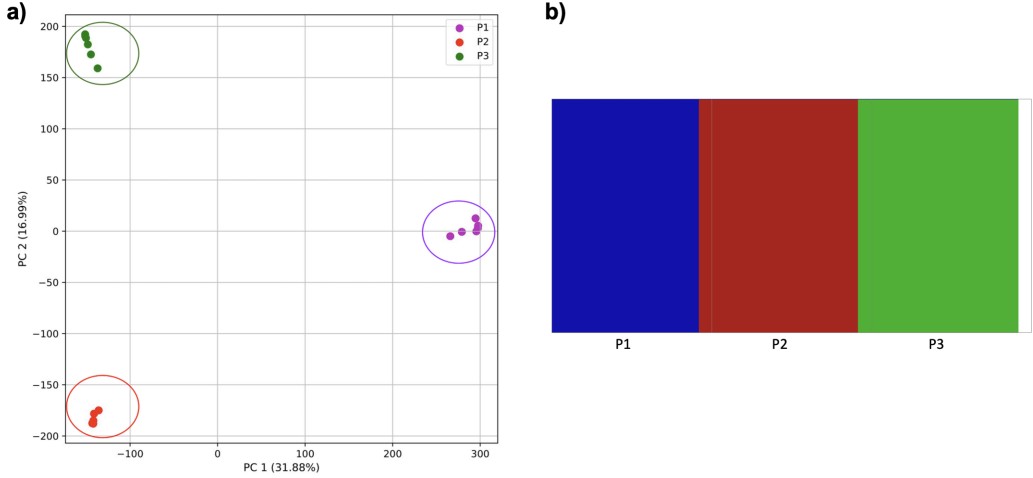

**Figure 8** **Comparison of population stratification approaches using simulated data with reduced time to population establishment.** Comparison of population stratification approaches three simulated populations loosely based on the human out-of-Africa population model of *Gravel et al. (2011)* as in Fig. 7; however, with reduced time between the establishment of the third population and sampling using simulations. (A) PCA generated using k-mer frequencies from samples. Samples are colored by population. K-means algorithm accurately identified three clusters present in the data (circled). (B) The same three populations accurately identified from SNPs using *fastStructure*. Samples are colored by the assigned population.

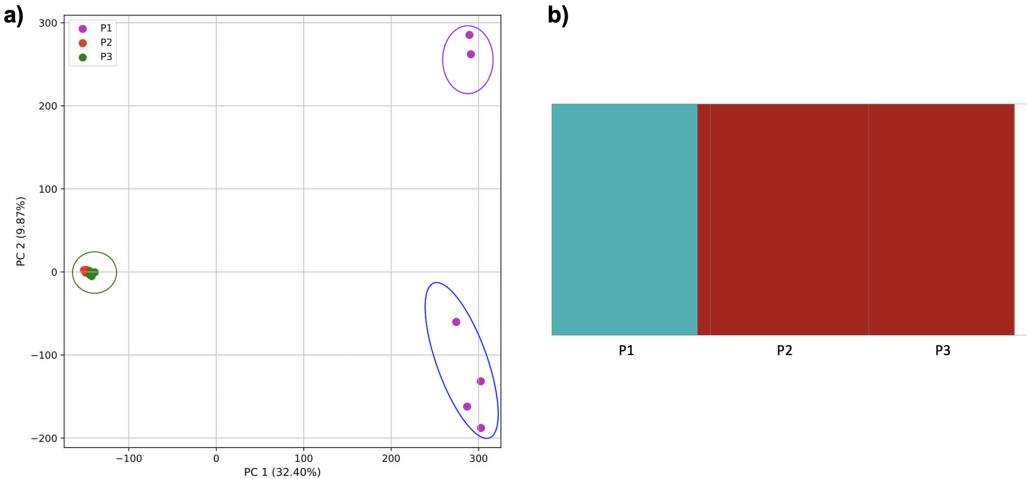

**Figure 9** **Comparison of population stratification approaches using simulated data with exponential growth of the populations after the establishment.** Comparison of population stratification approaches using three simulated populations loosely based on the human out-of-Africa population model of *Gravel et al. (2011)*, with exponential growth in populations 2 and 3 following establishment. (A) PCA generated using k-mer frequencies from samples. Samples are colored by population. K-means algorithm inaccurately identified three clusters present in the data (circled), with one including populations 2 and 3 combined, and the other separating two groups from population 1. (B) Two populations identified from SNPs using *fastStructure*. *fastStructure* assigned population 2 (pop2) and population 3 (pop3), to the same population. Samples are colored by the assigned population while simulated populations are identified along the x axis.

from population 3; cluster 2 consisted of 1 from population 1, 2 from population 2, and 2 from population 3; cluster 3 consisted of 1 from population 2 and 2 from population 3.

## Population structure from k-mer presence alone

For additional comparison with our PCA approach, we used *mash* to estimate population structure from the human samples. Monophyly of the superpopulation groups was observed on the unrooted tree for various parameters of k-mer length ($k$) and sketch size ($s$) (Figs. SA19–SA22). Specifically, the trend of accurate grouping by population was observed with shorter k-mer length and higher sketch size, and conversely lower sketch size and longer k-mer length produced trees with monophyletic groupings of samples by population (Fig. 15).

While we saw a general trend of improved accuracy with low k-mer length and higher sketch size, and conversely longer k-mers and smaller sketch size, we saw exceptions in the accuracy trend for various parameters. For example, k-mer length 24 and sketch size 3,000 produced accurate results, however, the accuracy dropped with setting $k = 24$ and $s = 5,000$ and 7,000, and the accuracy picked up again when $k = 24$ and $s = 8,000$–30,000. While this test case allowed us to compare results to the ground truth, it is difficult to select *a priori* the parameters that produce accurate results.

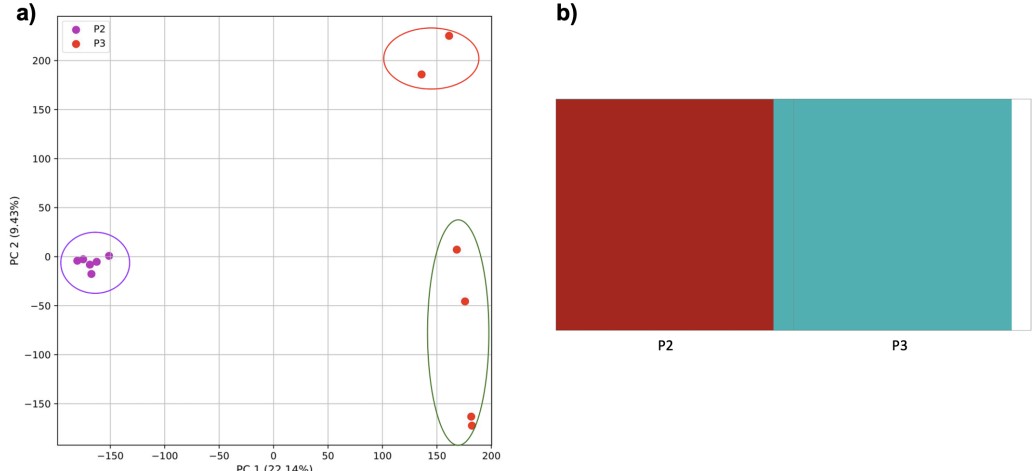

**Figure 10** **Comparison of population stratification approaches using simulated data reflecting the effect of the exponential growth in populations.** Comparison of population stratification approaches using the simulated samples from the two populations clustered in Fig. 9. Because analysis of all three populations separated grouped populations 2 and 3 together, these populations were analyzed separately to determine whether they could be separated from each other when population 1 was excluded. (A) PCA generated using k-mer frequencies from samples. Samples are colored by population. K-means algorithm inaccurately identified three clusters present in the data (circled). (B) Two populations accurately identified from SNPs using *fastStructure*. Samples are colored by the assigned population.

## DISCUSSION

In this work, we showed that population structure can be detected from k-mer frequencies using PCA and K-means clustering. We were able to assign samples to populations in a way that was comparable to prior work and simulations that used SNPs. PCA with K-means using k-mer frequencies was able to accurately assign samples to expected populations even in the scenarios where *fastStructure* did not differentiate subpopulations due to presence of high genetic admixture. Application of PCA to vectors of k-mer frequencies reduces the dimensionality of the data, which makes the dataset manageable for the following step of applying a clustering algorithm to detect structure in the dataset. We suggest that k-mer frequencies may be easier to calculate compared to accurate genotypes and allow the estimation of population structure hierarchically.

### Population identification

We hypothesize that our initial observation of two larger clusters in the data from the five human superpopulations (Africa and all others) reflects the effects of the high genetic variation and population-specific alleles in the African populations due to humans originating in Africa, combined with the bottlenecks and subsequent population expansion of other populations (*Campbell & Tishkoff, 2008*; *Gravel et al., 2011*). While similar substructure was not found by the 1000 Genomes Project when using SNPs, it is accurate that non-African populations share common ancestry with each other more recently than they do with African individuals. We observed that simulations produced similar differentiation in the results from the three-population model with

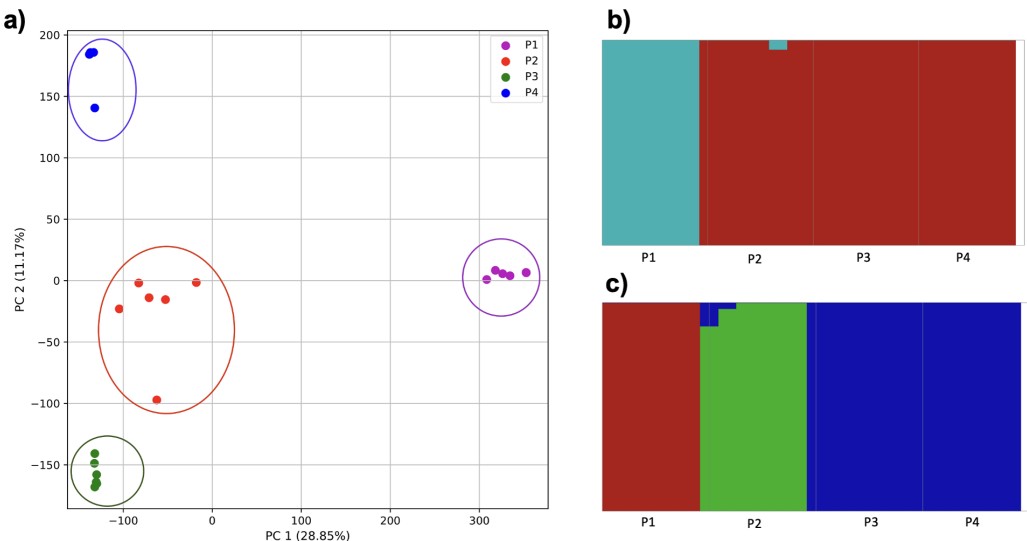

**Figure 11  Comparison of population stratification approaches using simulated data reflecting the effect of the mixed origin of a population.** Comparison of population stratification approaches using four simulated populations, where 1–3 are loosely based on the human out-of-Africa population model of *Gravel et al. (2011)*, and population 4 originated as a mixture of populations 2 and 3. (A) PCA generated using k-mer frequencies from samples. Samples are colored by population. K-means algorithm identified either three or four clusters (based on the scree plot) present in the data (circled). (B) *fastStructure* identified either two or three clusters from SNPs. (C) In the two-population case, *fastStructure* assigned population 2 (pop2), population 3 (pop3), and population 4 (pop4), to the same population. In the three-population case, *fastStructure* assigned population 3 (pop3) and population 4 (pop4), to the same population. Samples are colored by the assigned population.

population expansions, but not in the constant-size model, confirming that an out-of-Africa model with subsequent population expansion may produce the observations from the empirical data. Alternatively, *Kulohoma (2018)* found structure within the LWK samples corresponding to outlier individuals, who may be from different tribes, which could also explain differentiation within the African samples.

Excluding the African superpopulation, k-mer-based PCA grouped the samples by their expected superpopulation, with the majority of variation among superpopulations. This result was consistent with analyses based on SNPs (*The 1000 Genomes Project Consortium et al., 2015*) by the 1000 Genomes Project. PCA analysis of individual superpopulations similarly differentiates populations as expected in prior work. These results are consistent with the results from our simulations, in which populations were easily differentiated by our approach.

For populations identified as having multiple origins using SNPs, k-mer-based PCA places samples in between the two populations representing the two ancestral populations. The continuum in the placement of samples of multiple origins likely corresponds to the fractional membership of samples as determined by the 1000 Human Genomes Project. Specifically, given two populations, additional samples with origin similar to both are placed on the coordinates along edges joining the centers of the established populations (*Patterson, Price & Reich, 2006*). While it is not surprising that these samples are not

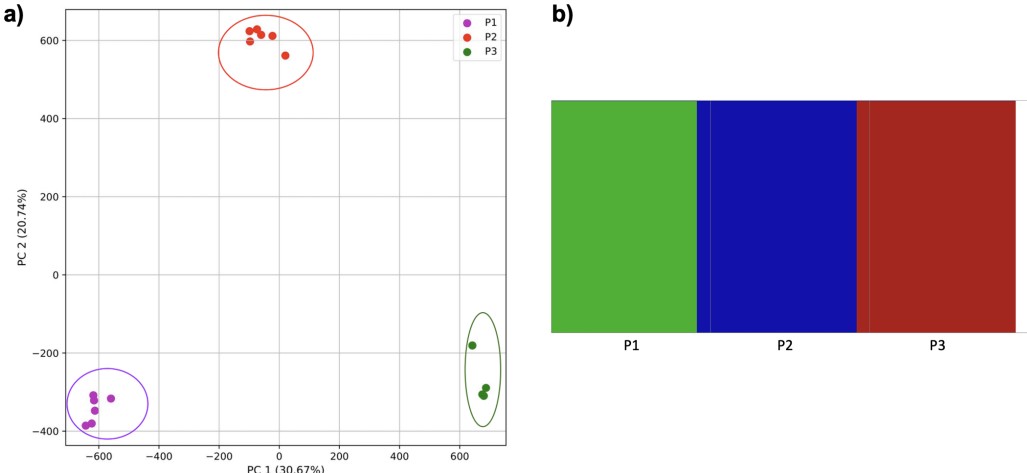

**Figure 12  Comparison of population stratification approaches with simulated data reflecting effect of hybrid-origin population isolation.** Starting with two larger and one smaller population, the smaller population initially experienced migration from the larger, and this was then reduced. (A) PCA generated using k-mer frequencies from samples. Samples are colored by population. K-means algorithm accurately identified three clusters present in the data (circled). (B) Three populations identified from SNPs using *fastStructure*. Samples are colored by the assigned population.

accurately assigned to a third population, as this same result was found by the 1000 Genomes Project (*The 1000 Genomes Project Consortium et al., 2015*), three populations were identified by K-means and the third intermediate population was largely comprised of the expected mixed samples.

## K-mers *versus* marker genotypes

Importantly, k-mer frequencies can be simpler to count and process when compared to using marker genotypes (*e.g.*, SNPs). Thus, the k-mer-based approach has particular potential to be useful for non-model organisms (*Russell et al., 2017*). Such non-model organisms do not have established SNP panels or information on allele frequencies from which population structure can be identified using model-based approaches and marker genotype data. In organisms without a reference genome (*Bendaoud et al., 2022*), a typical way to discover SNPs to analyze variation is through restriction digests and several computational pipelines (*Puritz, Hollenbeck & Gold, 2014*). On the other hand, generation of k-mer profiles from genome sequences, even of non-model organisms, is a straightforward task where subsequences of length k are counted along the genome and no markers have to be identified. While sequencing whole genomes can be more costly than RADseq, whole genome sequencing promotes reuse of data and is becoming more cost effective. Therefore, the method described in this paper provides a functional alternative to model-based approaches where genome data lacks information on SNPs and provided the goal of the analysis is to only identify the population structure present in the dataset. It may also act as a complementary approach to analyses where larger numbers of individuals were sequenced for smaller allele panels.
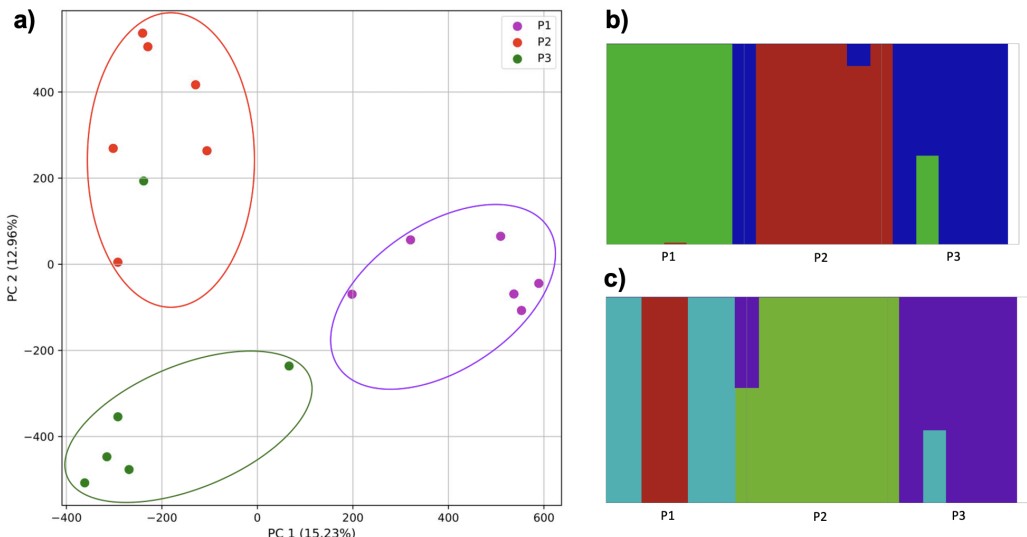

**Figure 13** **Comparison of population stratification approaches with simulated data reflecting effect of increase in population migration.** Starting with two larger and one smaller population, the smaller population initially experienced migration from the larger, and this was then reduced (although to a lesser degree than Fig. 12). (A) PCA generated using k-mer frequencies. Samples are colored by population. K-means algorithm identified three clusters present in the data (circled), which corresponded closely, but not exactly, to expectation based on sampling origin. (B) In the three-population case, *fastStructure* mostly assigned populations correctly with one individual assigned incorrectly as in the PCA. (C) In the four-population case, *fastStructure* primary assigned individuals correctly, with one exception; additional population 1 was divided into two clusters. Samples are colored by the assigned population.

When analyzing genotype data, a number of preprocessing steps to evaluate data quality are necessary (*Pritchard, Stephens & Donnelly, 2000*; *Alhusain & Hafez, 2018*), as population structure estimation using genetic markers is susceptible to genotyping errors (*Pompanon et al., 2005*). This evaluation includes an assessment of SNP call rates, minor allele frequencies (MAFs), verification of the HWE assumptions, and relatedness between individuals. Additionally, the identification of ancestry informative markers, which constitute a minimal number of markers needed to obtain population structure, is necessary to ensure the accuracy of the results (*Alhusain & Hafez, 2018*). In contrast, k-mer frequencies can be viewed as summary statistics of a genome resulting from SNPs. While errors in k-mer counts occur due to sequencing or genotyping errors, these can easily be identified as k-mers of low frequency. The overwhelming majority of k-mers of frequency 1 are not found in a genome and thus are most likely due to sequencing errors and therefore can easily be discarded (*Melsted & Pritchard, 2011*). While in this analysis we use the available reference genome for alignment to produce a genome for each sample, k-mer frequencies should not be significantly affected by using a draft genome assembly.

Model-based population analysis methods can also be limited by inadequate sample sizes and the number of markers analyzed (*Lawson, Van Dorp & Falush, 2018*). These methods depend on an estimation of allele frequency that is sensitive to small samples (*Gao & Starmer, 2008*; *Porras-Hurtado et al., 2013*). In contrast, our PCA-based k-mer frequency

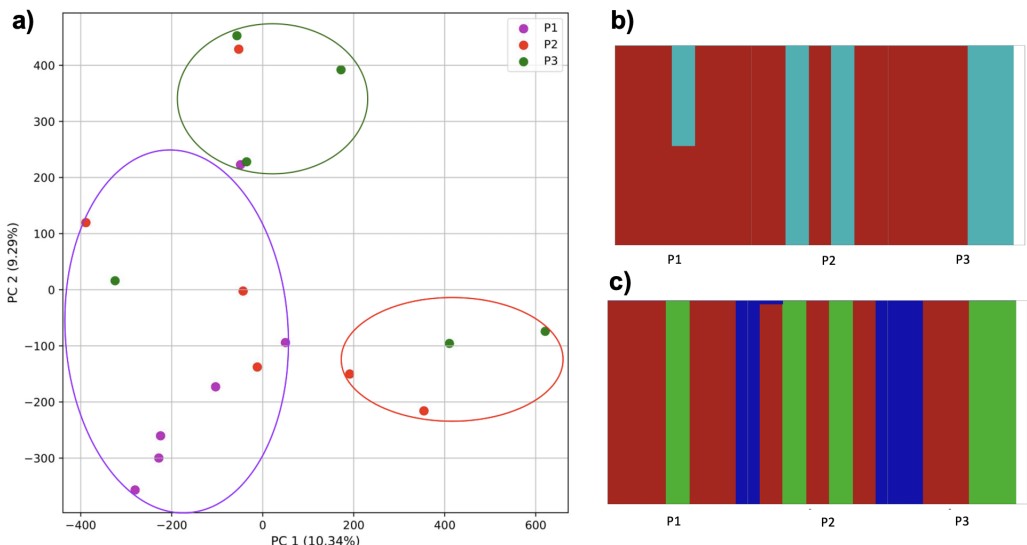

**Figure 14** **Comparison of population stratification approaches from simulated data reflecting effect of further increase in population migration.** Starting with two larger and one smaller population, the smaller population initially experienced migration from the larger, and this was then reduced (although to a lesser degree than Fig. 13). (A) PCA generated using k-mer frequencies from samples. Samples are colored by population. K-means algorithm identified three clusters present in the data (circled), but at this level of migration they did not correspond to sampling location. Under two- (B) and three-population (C) scenarios, admixed populations did not correspond to the original simulations using SNPs and *fastStructure*.

approach does not depend on allele frequency estimation and thus is not affected by sample size (*Alhusain & Hafez, 2018*). We were able to produce accurate results with as few as six samples per population.

The PCA-based approach also has the advantage of operating without a preset number of populations and no modeling assumption requirements (*Patterson, Price & Reich, 2006*; *Lee, Abdool & Huang, 2009*). This makes the analysis of population structure using a non-model-based approach an appealing choice. The application of PCA and K-means to k-mer profiles of genomes makes it easy to detect a number of populations (clusters) present in the dataset, which is a major parameter in the model-based method that is required to be set in advance.

However, it is important to recognize that the k-mer and SNP-based approaches may deliver different types of information. While a more complex but sophisticated method such as *STRUCTURE* describes the population structure by probabilistic assignment to classes, the PCA combined with K-means, provides a graphical and quantitative representation of population structure along axes of variation in the dataset. Thus, the goal of the investigation of population structure should be taken into consideration when deciding which approach is more applicable for the analysis.

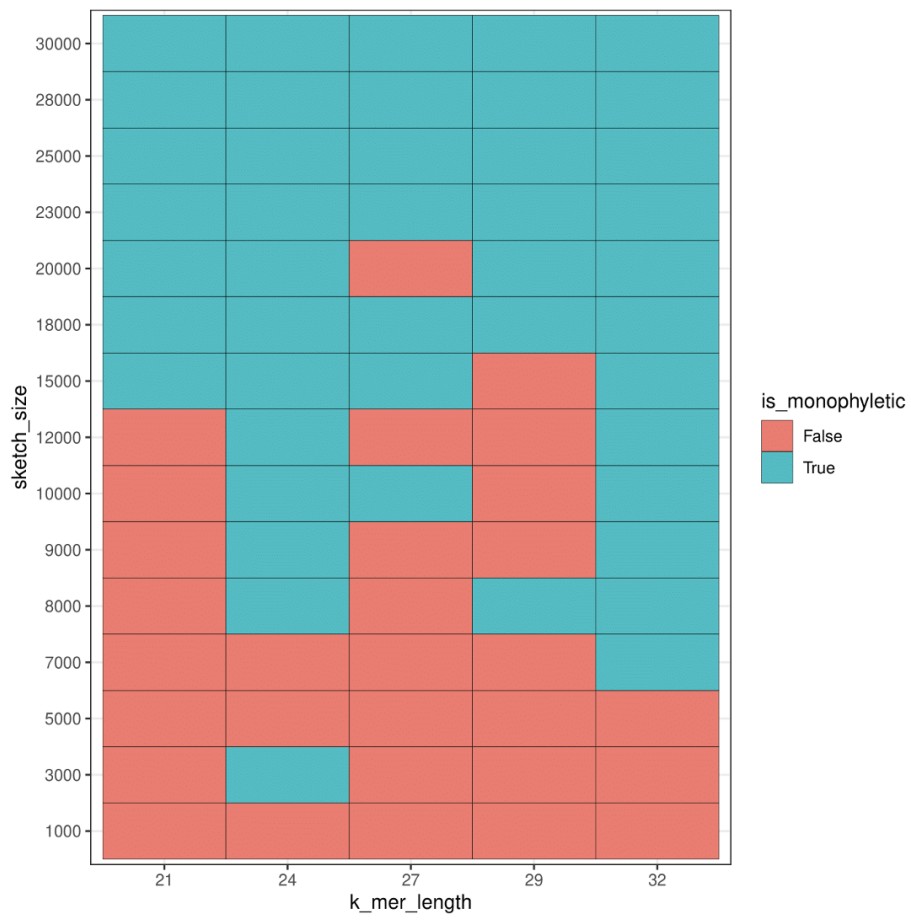

**Figure 15** **Heatmap plot showing variation in the ability of *mash* to detect monophyly of human superpopulations.** Phylogenies were built from pairwise mash distances for different k-mer length and sketch size parameters.

## Consideration of computational efficiency

The PCA-based approach is computationally efficient and can handle genomic marker data for thousands of individuals (*Paschou et al., 2007*). PCA also demonstrated efficiency when applied to k-mer frequencies in our study. However, the k-mer approach has some limitations. While it is effective for analyzing datasets with a limited number of samples, the large memory burden associated with k-mer analysis poses a significant challenge when applied to larger datasets. The memory requirements increase with k-mer length, as does the computational power needed to calculate distances between sequences based on k-mer frequencies. Additionally, shorter k-mers tend to be less informative, whereas longer k-mers offer greater specificity but demand more resources. This creates a tradeoff between specificity, computational efficiency, and memory requirements. Approaches such as down sampling (sketching) or compressing the k-mer space may help address these limitations and improve the feasibility of k-mer analysis for larger datasets.

### Identification of population structure based on the number of shared k-mers

*mash's* identification of all five superpopulations, suggests that *mash* has an adequate amount of sensitivity to differentiate samples by superpopulation even with the presence of samples with greater variation (AFR). However, grouping individuals into populations based on k-mer presence using *mash* distances was more difficult than using PCA and k-mer frequencies. *mash* produced accurate groupings (samples were placed on the tree according to the superpopulation) only for particular combinations of k-mer length and sketch size, and accuracy was not necessarily predictable. We were able to identify the k-mer length that produced accurate results by checking with the results produced by the 1000 Genome Project; however, finding the right parameter for k-mer length would be hard without knowing the correct population structure in advance. Thus, while *mash* is a robust tool for identifying genome relationships on a species level (*Zielezinski et al., 2017*), on a population level, in comparison to the PCA approach, *mash* showed less viable performance for differentiating populations due to its high sensitivity to parameter selection which is unknown in advance.

### Whole genome sequences *versus* raw sequencing reads

Our study utilized k-mers derived from whole genome sequences (WGS) to enhance accuracy and minimize the impact of sequencing errors and artifacts. We used this approach to ensure that k-mers may be used successfully to determine population structure. However, an optional approach, especially for non-model organisms without an assembled genome, may be to use k-mers directly from raw sequencing reads (*Rahman et al., 2018*). Raw reads are inherently error-prone while having variation in read coverage that affects k-mer frequencies; however, it may be possible to address these challenges through filtering (*e.g.*, removing rare k-mers) and trimming techniques, with tools like *mash* effectively mitigating errors and contamination. This direct approach not only avoids the need for a reference genome, but bypasses the alignment step, potentially streamlining the analysis and improving computational efficiency. Future work should consider the steps necessary to match the effectiveness of genome-based k-mer analysis that we observe here while using read-based k-mer counts.

## CONCLUSIONS

In sum, principal component analysis together with K-means clustering appears to successfully identify population structure based on the k-mer frequencies present in genomes. This approach is robust in differentiating samples at the superpopulation and population levels. Notably this approach differentiated samples hierarchically, and PCA was able to discern the population signal in samples of multiple origins within a single superpopulation. These results are comparable to model-based approaches that identify populations using genotypes, and which provide information on fractional membership of a sample to a population. However, using k-mer frequencies does not depend on genetic assumptions or the process of marker selection curation. In contrast, the method using k-mer presence to group samples lacked sensitivity to consistently identify populations.

With the increasing availability of whole-genome data, we anticipate that the use of k-mer frequencies combined with PCA and K-means clustering can provide information that is complementary to marker-based work in population structure investigations. This genome-wide pattern-based approach provides an initial glimpse at an interesting alternate source of information that may be further investigated through more complex simulations and empirical work. While our approach still requires availability of a reference genome, as many current approaches do, a k-mer-based method using PCA with K-means offers potential for a simplified procedure for population stratification. It also shows promise for enhancing the process of population structuring and assignment, even in the presence of high genetic admixture.

## ACKNOWLEDGEMENTS

We thank members of the Schwartz lab and the anonymous reviewers for valuable comments and suggestions. The High-Performance Research Computing (HPC) facility at the University of Rhode Island is acknowledged for providing the computational resources for the analyses in this manuscript. Additionally, we thank Dr. Kevin Bryan, the HPC systems manager, for invaluable help with software installation on the cluster and GPU operations management.

### Funding

This research was funded by a grant to Rachel S. Schwartz from the National Science Foundation (DBI-1942273) and by the USDA National Institute of Food and Agriculture, Hatch project accession no. 1017848. The funders had no role in study design, data collection and analysis, decision to publish, or preparation of the manuscript.

### Grant Disclosures

The following grant information was disclosed by the authors:
National Science Foundation: DBI-1942273.
USDA National Institute of Food and Agriculture, Hatch project: no. 1017848.

### Competing Interests

The authors declare there are no competing interests.

### Author Contributions

- Yana Hrytsenko conceived and designed the experiments, performed the experiments, analyzed the data, prepared figures and/or tables, authored or reviewed drafts of the article, and approved the final draft.
- Noah M. Daniels conceived and designed the experiments, authored or reviewed drafts of the article, and approved the final draft.
- Rachel S. Schwartz conceived and designed the experiments, performed the experiments, analyzed the data, prepared figures and/or tables, authored or reviewed drafts of the article, and approved the final draft.

## DNA Deposition

The following information was supplied regarding the deposition of DNA sequences:

The open-access genome data used in this work is available at the 1000 Human Genomes Project. Other genome data was generated using simulations: ERR3239277, ERR3239283, ERR3239290, ERR3239295, ERR3239484, ERR3239489, ERR3239496, ERR3239501, ERR3239503, ERR3239510, ERR3239515, ERR3239522, ERR3239559, ERR3239561, ERR3239566, ERR3239578, ERR3239583, ERR3239604, ERR3239684, ERR3239689, ERR3239691, ERR3239696, ERR3239710, ERR3239723, ERR3239834, ERR3239839, ERR3239846, ERR3239851, ERR3239853, ERR3239857, ERR3240226, ERR3240232, ERR3240234, ERR3240239, ERR3240246, ERR3240251, ERR3241985, ERR3241992, ERR3242000, ERR3242128, ERR3242140, ERR3242142, ERR3242201, ERR3242216, ERR3242239, ERR3242246, ERR3242260, ERR3242265, ERR3242911, ERR3242915, ERR3242922, ERR3242927, ERR3242934, ERR3242936.

## Data Availability

The code to download the data from the 1000 Human Genomes Project, run the analysis, generate simulated data, and complete the analysis is available in GitHub and Zenodo:

- https://github.com/YanaHrytsenko/determining_population_structure_from_k-mer_frequencies

- Yana Hrytsenko. (2024). YanaHrytsenko/determining_population_structure_from_k-mer_frequencies: v1.0.0 (v1.0.0). Zenodo. https://doi.org/10.5281/zenodo.13823215

- https://github.com/SchwartzLabURI/kmer_freq_popstructure_sims

- rachelss, & Yana Hrytsenko. (2025). SchwartzLabURI/kmer_freq_popstructure_sims: v1.0.0 (PeerJ_publication). Zenodo. https://doi.org/10.5281/zenodo.14679764

## Supplemental Information

Supplemental information for this article can be found online at http://dx.doi.org/10.7717/peerj.18939#supplemental-information.

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
