# Peer review of "Determining population structure from k-mer frequencies"

_PeerJ, doi:10.7717/peerj.18939_

## Round 0.1 · original submission · Major Revisions

Although all comments should be responded to, authors should pay special attention to reviews by Reviewer 4 and Reviewer 1. Note in particular that both have observed that the claim that the proposed approach is more computationally efficient than others must be supported by clear evidence.

Reviewer 1 ·

Basic reporting

- Several software packages were utilized in this study, including ape, fastStructure, ggplot, Jellyfish, mash, Python, scikit-learn, and SLiM; however, the specific versions were not reported. Additionally, it remains unclear which implementation of the K-means algorithm the authors used. To enhance the clarity and reproducibility of the study, the authors should specify the software versions and the source of the K-means algorithm.
- Since the authors used fastStructure rather than Structure, I recommend changing the section title in line 323 from "Comparison of simulated data results using STRUCTURE" to "Comparison of simulated data results using fastSTRUCTURE".
- The authors should review reference 74 to ensure they have cited the correct source, rather than inadvertently copying a notice from the SLiM manual.
- The authors should also carefully review several references marked as preprints, as some of these works have since been published in academic journals, books, or conferences, and should no longer be classified as preprints.

Experimental design

No comment

Validity of the findings

- In line 121, the authors claim that "existing methods to identify population structure are computationally intensive," but they do not specify in what way these methods are computationally demanding. For example, FlashPCA2 (Abraham et al. 2017) can perform PCA on biobank-scale data using 2 GB for 1,000,000 individuals and 100,000 SNPs in under 12 hours with a single core. In contrast, the authors' method requires more than 2 hours in wall-clock time with 36 cores (equivalent to over 72 hours in CPU time) and at least 250 GB of memory (Lines 457–464). If the authors believe their approach is more computationally efficient, they should provide evidence to support this claim. Otherwise, they should discuss the limitations of their method, as it appears to be suitable only for analyzing data with a limited number of samples.

Additional comments

- In this GitHub repository https://github.com/YanaHrytsenko/determining_population_structure_from_k-mer_frequencies, I recommend that the authors list all Python dependencies along with their respective versions to improve the reproducibility of their work.
- In this GitHub repository https://github.com/SchwartzLabURI/kmer_freq_popstructure_sims, I recommend that the authors list all Python dependencies along with their respective versions to improve the reproducibility of their work. Moreover, since this repository contains code from https://github.com/YanaHrytsenko/determining_population_structure_from_k-mer_frequencies, which is under the MIT License, the authors should retain the MIT License for any code under that license to comply with its requirements. Additionally, the authors should refactor the repository to align its style with Yana Hrytsenko's repository by organizing scripts for different steps into separate folders and creating a dedicated folder for all SLiM simulation scripts. They should also rewrite the README in markdown format to improve readability and clarity for users.

Cite this review as

Reviewer 2 ·

Basic reporting

The title of the article in suitable.
The English is suitable and good for scholarly articles.
The references are sufficient and recent and also relevant to the present study.
The graphs are figures are self explanatory and of sufficient resolution.
The headings and subheadings are organized properly.

Experimental design

The abstract of the article is comprehensive. The introduction and methodology is appropriate and adequate. The result and discussion has been presented nicely but the discussion part should be little bit elaborated and the conclusion of the manuscript effectively summarizes the key findings of the study and also the authors have successfully highlighted the broader implications of their work, especially in relation to specific field, which enhances the relevance of their research.

Validity of the findings

The study employed a suitable methodology for addressing the research question, and the choice of analytical tools appears appropriate given the nature of the data. The methods used are clearly described, ensuring transparency and reproducibility of the study. In terms of results, the authors have presented their findings with clarity, supported by appropriate statistical analysis. The discussion section effectively correlates the findings with the existing body of literature, highlighting both the novel contributions of the study and its alignment with previously established research. The conclusions drawn are consistent with the data.

Additional comments

Please write the full form before using abbreviations.
Ensure every figure/table is properly referenced in the text immediately after the data is presented.
Ensure references are as per the journal's guidelines.

Cite this review as

Reviewer 3 ·

Basic reporting

No comment

Experimental design

No comment

Validity of the findings

No comment

Additional comments

This paper describes an approach to determine population structure directly from short-read sequencing k-mers rather than from genotypes inferred from short-read sequencing mapped to a reference genome.
The approaches used, including principal component analysis and k-means clustering, are standard for genotyping data, and therefore their application to k-mer frequencies is well-justified. The analyses are well-justified and described, and applied to real data and simulated data.
The only major comment that the authors should address concerns care in the choice of language when describing results and samples in human population genetics. Use of terms such as “ancestral” is OK in the context of admixture, as described in the section starting at line 269, but care must be taken not to imply that these groupings represent distinct “original human groups”. For example, line 269 “multiple ancestral originals (sic)” and line 173 “humans consist of five superpopulations”. These five superpopulations are sampling groups roughly corresponding to geography (but not always) and don’t have much biological meaning – the “lumpiness” derives mostly from the sampling rather than discrete boundaries in genetic diversity. Even the large difference between sub-Saharan Africa and the rest of the globe is due to the rest of the globe being a sub-sample of SSA diversity. The authors should carefully review these descriptions throughout the paper and adjust as appropriate to prevent misinterpretation.

Cite this review as

Reviewer 4 ·

Basic reporting

Article is pretty clear and self-contained.

Experimental design

This manuscript asks whether population structure can be investigated by counting k-mers (small subsequences of larger sequences) instead of SNPs called from read alignments to a reference genome. This is an interesting and practically relevant question, especially in non-model systems that lack many reference genomes or with very large, heterozygous, or polyploid genomes. I also believe it’s an open question how much reference bias and pangenomic variation distorts our view of population structure. The authors investigate the ability of k-mers to determine population structure using data from 48 human genomes and with simulations. Overall, I am very enthusiastic about the question posed and am very excited to see this question explored with simulations.

Validity of the findings

One of the main perks of studying k-mers is that they can be counted in raw sequencing reads without doing alignment. This is how k-mers are typically analyzed and it allows them to pick up on non-reference variation and avoid computationally intensive alignments. However, unless I am misunderstanding something, the way the authors count k-mers is by first aligning reads to a reference genome (lines 190 – 199). The authors count k-mers by using previously generated cram files (which are generated by aligning sequencing reads to a reference genome), calling SNPs, and then using the SNP calls combined with the human reference genome to build a representation of a genome for each of their samples in fasta format. They then count k-mers on these fasta files. This seems very unusual to me and defeats part of the reason why one might want to analyze k-mers in the first place instead of the standard SNP calling approach. This also might explain why later in their methods the authors do not find many 21-mers with a frequency greater than 1 in their samples (lines 302 - 203). This is to be expected if the authors counted 21-mers in genomes instead of sequencing reads. We would expect 21 bp sequences to often map to unique genomic locations. This is the same logic for why people design PCR primers to be in the range of >20 bp – with longer primers, the chance of them binding to off target sites is very small. In contrast, we would expect the same 21-mer to occur many times in a set of raw-sequencing reads because a deeply sequenced sample will include reads for every position in the genome many times. Later in the study, the others also exclude k-mers with a frequency of 1 as potential sequencing errors. This is a common practice to exclude low frequency k-mers when working with sequencing reads. However, that logic would not apply here because the authors are only working with k-mers that are derived from (presumably) high confidence alignments to a reference genome. To me, it would make much more sense to count k-mers in the original sequencing reads that were used to generate the cram files and use those counts to study population structure. In summary, the choice to count k-mers in reference genomes (fasta files) instead of sequencing reads (fastq files) must be explained and justified.

Furthermore, for lines 121 – 123 the authors say in their intro that “existing methods to identify population structure are computationally intensive. The number of available markers grows as the number of samples included in the analysis increases”. While I agree that inferring population structure with SNPs can require a good amount of computing resources, k-mers by themselves do not necessarily solve this issue. While a typical SNP analysis includes a few million SNPs, a typical k-mer analysis can include many millions of k-mers and the authors needed over 200 Gb of RAM for calculating k-mer-based PCA (lines 456-464). There should be some sort of acknowledgement that this large memory burden is a limitation of k-mer analysis, though approaches for either downsampling (sketching) or compressing the k-mer space can potentially remedy this issue.

Additional comments

Unless I am missing it, I don’t think the terms “superpopulation” , PCA, and GWAS are ever defined. Though these are very common terms in population genetics, it would be good to redefine them here so that the paper is accessible to a broader audience.

For figures 1 – 6, I believe only PCAs on k-mer datasets are shown. It could be nice to show the same PCA for SNP data side-by-side for easy comparison with the k-mer PCA, similar to how figures 7 and onward show k-mer and SNP plots side-by-side.

It could be helpful if this study included some basic control simulations to make the results easier to understand and interpret. Perhaps I am missing it, but my expectation was that this manuscript would begin by simulating the simplest possible evolutionary scenario: a single neutrally evolving population of constant size. This population would be expected to have no structure, and they could apply both the k-mer and snp-based approaches and see if they get an optimal cluster value of 1 in both cases. They could then build up to a two-population model with varying levels of migration, then a three-population model and so on. Instead, the first model is a three-population model. Not having these more basic benchmarking simulations seems like a missed opportunity, though I recognize adding them at this stage would be a lot of additional work.

I appreciated the opportunity to review this manuscript.

Cite this review as

---

## Round 0.2 · Minor Revisions

Reviewer #4 provided additional comments, and requests elaboration of your discussion based on the points raised. I agree that this is needed.

Reviewer 1 ·

Basic reporting

The authors have addressed my concerns, I have no further comments.

Experimental design

No comment.

Validity of the findings

The authors have addressed my concerns, I have no further comments.

Additional comments

The authors have addressed my concerns, I have no further comments.

Cite this review as

Reviewer 4 ·

Basic reporting

No comment.

Experimental design

I appreciate the response to my comment about counting k-mers in genomes vs sequencing reads and the clarifications made in the manuscript text. I do remain somewhat unconvinced by the justification provided. I agree that raw sequencing reads contain noise and sequencing error and alignment can address some of these issues. However, as the manuscript acknowledges, sequencing errors could be largely addressed by excluding k-mers that only occur a few times in a set of reads. The raw sequencing reads could also be filtered and trimmed before counting k-mers using read trimming tools. And contamination could be addressed with mash (one of the tools in the manuscript), which allows one to rapidly screen sequencing reads against a reference database of microbial genomes. Contributions from repetitive regions could be better accounted for with metrics like bray-curtis or cosine dissimilarity, which are similar to the Jaccard index but account for copy number differences between k-mers instead of just encoding k-mer presence/absence.

Comparing SNPs and k-mers that are both derived from alignments seems like two slightly different ways of encoding the same information because each biallelic SNP will be tagged by about 2 times k k-mers. Though one exception to this would be that a single k-mer can tag multiple SNPs at once if the SNPs are within k bases of each other (in this situation the k-mers essentially represent a small haplotype).

Rahman et al. 2018, which is cited in the manuscript and seems to be a primary motivation for the research, also uses k-mers to investigate population structure in humans, but by counting k-mers in raw sequencing reads - not genome assemblies. Overall, this makes analysis of k-mers more meaningfully different from analysis of SNPs because it allows one to skip computationally demanding read alignment.

At the end of the day, I think that the manuscript's methods are still valid. K-mer-based genetic distance measures between genome assemblies are still valid, just as they are for sequencing reads. I just think methods based on k-mers from sequencing reads would be of wider utility and would be a good avenue for future research. I think adding some sort of acknowledgement to the discussion on this point would be helpful (unless this was already added, in which case I apologize for missing it).

Validity of the findings

No comment.

Additional comments

No comment.

Cite this review as

---

## Round 0.3 · Minor Revisions

The new section adequately responds the comment by R4. However, it seems to contain one small typo that needs correction. You wrote: "an optimal approach" (line 807); I believe you meant "an optional approach". If my interpretation is correct, please revise and resubmit. If not, please justify why a read-based approach would be optimal.

---

## Round 0.4 · accepted · Accept

Thanks for the revised version. Your manuscript is now ready for publication.